# Comprehensive Clinical Profile of *Amanita exitialis* Poisoning: Integrating Toxin Detection and Autopsy Pathology

**DOI:** 10.3390/toxins17120576

**Published:** 2025-11-29

**Authors:** Chong-Gui Chen, Ping Xu, Ji-Pin Li, Xiao-Li Bi, Qun-Mei Yao, Cheng-Min Yu, Yan Tang, Cheng-Ye Sun, Zhi-Jun Wu, Jia-Ju Zhong, Hai-Ying Wu

**Affiliations:** 1Clinical Research Center for Mushroom Poisoning, The People’s Hospital of Chuxiong Yi Autonomous Prefecture, Chuxiong 675000, China; ccg09242025@163.com (C.-G.C.); xupingree@163.com (P.X.); lijipin2025@163.com (J.-P.L.); bxl19987217574@163.com (X.-L.B.); yaoqunmei1970@163.com (Q.-M.Y.); ynycm113@163.com (C.-M.Y.); 2College of Pharmacy, Dali University, Dali 671000, China; 3Department of Emergency Medicine, The People’s Hospital of Chuxiong Yi Autonomous Prefecture, Chuxiong 675000, China; tangyanqs2046@163.com (Y.T.); wuzj_09@126.com (Z.-J.W.); 4National Institute of Occupational Health and Poison Control, Chinese Center for Disease Control and Prevention, Beijing 100050, China; suncy@chinacdc.cn; 5Department of Emergency and Intensive Care Unit, The First Affiliated Hospital of Kunming Medical University, Kunming 650032, China

**Keywords:** *Amanita exitialis*, mushroom poisoning, amatoxins, toxin detection, autopsy pathology, clinical characteristics

## Abstract

*Amanita exitialis* is a lethal mushroom species found in southern China. Its amatoxins can cause acute liver injury with a high case-fatality rate. However, reports combining toxin detection in clinical specimens with autopsy pathology remain limited. We conducted a retrospective analysis of *A. exitialis* poisoning events treated at Chuxiong Yi Autonomous Prefecture People’s Hospital from 2019 to 2024. Toxins were measured in collected mushrooms, patient blood, and urine. Clinical data included demographics, complications, laboratory parameters, and autopsy findings. Associations between a time-weighted urinary amatoxin exposure metric and laboratory indices were assessed. Ten poisoning incidents involving 27 individuals were identified, including five deaths. We collected 10 mushroom samples, 120 urine samples, and 108 blood samples. α-amanitin, β-amanitin, phallacidin, and phallisacin were detected in mushrooms and urine. The detection rates of α-AMA, β-AMA, PCD, and PSC in urine samples were 31.67%, 5.00%, 38.33%, and 49.17%, respectively. Only three blood samples tested positive for α-AMA. The time-weighted urinary amatoxin exposure metric was positively correlated with total bilirubin (TBIL), aspartate aminotransferase (AST), alanine aminotransferase (ALT), blood urea nitrogen (BUN), creatinine (Cr), creatine kinase (CK), creatine kinase isoenzymes (CK-MB), prothrombin time (PT), activated partial thromboplastin time (APTT), and international normalized ratio (INR). Early symptoms included nausea, vomiting, diarrhea, abdominal pain, and distention; later findings involved injury to the liver, kidneys, intestines, heart, and lungs. On the fourth day following ingestion, there was a marked increase in bilirubin levels and a concurrent decrease in liver enzymes, indicating severe damage to the hepatocytes. Platelet count, white blood cell count, hemoglobin, and red blood cell count decreased over time. Autopsies demonstrated hepatic, renal, and myocardial injury, gastrointestinal mucosal exfoliation, and multiorgan hemorrhage. In summary, *A. exitialis* poisoning is primarily characterized by liver damage, accompanied by injuries to the kidneys, myocardium, and intestines, as well as multiorgan hemorrhaging, which may lead to blood toxicity. The detection rate of toxins in urine samples is relatively high, and early urine toxin testing can help clarify the diagnosis and guide treatment.

## 1. Introduction

Mushroom poisoning is a public health concern in many countries, contributing to substantial morbidity and mortality. *Amanita* species containing amatoxins account for over 70% of fatalities [1,2]. In the Americas and Europe, the principal lethal species are *Amanita phalloides*, *Amanita verna*, and *Amanita virosa* [3,4,5,6,7]. In contrast, fatal mushroom poisoning in China is primarily caused by *Amanita exitialis*, *Amanita subjunquillea*, *Amanita pseudoporphyria*, *Amanita subpallidorosea*, and *Amanita rimosa* [8,9]. From 2019 to 2023, poisoning by *A. exitialis* resulted in 135 cases and 24 deaths, ranking as the leading cause of mortality among mushroom-poisoning species in China [2,8,9]. Taken together, these data indicate that *A. exitialis* is currently the most lethal species implicated in mushroom poisoning incidents.

Several case reports have documented fatalities resulting from *A. exitialis* poisoning. During 2014–2015, three incidents involving 10 patients in Yunnan Province, China, were reported by Sun [10], and four deaths due to fulminant hepatic failure were recorded. In 2019, a family outbreak involving five individuals in Yunnan Province was reported, in which two children were poisoned, and one child died from hepatic failure complicated by multiple organ failure [11]. In 2022, a series of 10 patients in Shenzhen, China, with acute liver injury, abdominal distension, and thrombocytopenia was described by Meng, with one death from liver failure [12]. Additionally, in 2022, a case of *A. exitialis* poisoning with acute liver injury and prominent pneumatosis intestinalis and hepatic portal venous gas was reported by Chen [13]. Although awareness of mushroom poisoning prevention has increased, the number of reports of mushroom poisoning has also risen. However, the quality of evidence and clinical data for these cases is inconsistent.

Considering the sporadic and complex nature of mushroom poisoning, current reports on *A. exitialis* poisoning are limited. Research on *A. exitialis* poisoning reveals significant shortcomings in detecting toxins in patients’ bodily fluids and conducting autopsy pathological analyses. This study aims to fill these gaps by conducting an in-depth analysis of toxin metabolic patterns in urine and pathological changes in organ tissues. The results not only bridge critical gaps in clinical–pathological research on amanitin poisoning but also provide crucial evidence for the development of antidotes and the optimization of therapeutic windows.

## 2. Results

### 2.1. Clinical Manifestations

All 10 collected mushroom specimens were identified as *A. exitialis*, and their sequence accession numbers were submitted to the NCBI database (Table 1). A total of 27 patients from 10 poisoning incidents were included, of whom 5 (19%) died. As shown in Table 2, 13 patients (48%) were male. Poisoning cases predominantly occurred between June and August. The mean age was 46.8 ± 22.1 years. The latency period (from ingestion to symptom onset) was 12.0 ± 5.0 h, the time from ingestion to hospital admission was 45.9 ± 8.6 h, and the hospital stay was 8.2 ± 5.8 days. All patients (100%) exhibited gastrointestinal symptoms, including nausea, vomiting, diarrhea, and abdominal pain. In addition, liver injury occurred in 21 patients (78%), myocardial injury in 24 (89%), Kidney injury in 17 (63%), thrombocytopenia in 15 (56%), coagulation dysfunction in 10 (37%), anemia in 13 (48%), and hyperlactatemia in 8 (30%). Compared to the survivors, there was a statistically significant difference in coagulation dysfunction and hyperlactatemia among the non-survivors. One patient developed severe abdominal distension on the fourth day after ingestion. Imaging (Figure 1) revealed widespread gas accumulation and distention of the small intestine, with some segments showing a “spring coil”-like appearance. This patient died on the fifth day following ingestion.

### 2.2. Laboratory Examination

Compared with the Survivors group, the Non-Survivors group exhibited significantly higher levels of total bilirubin (TBIL), alanine aminotransferase (ALT), aspartate aminotransferase (AST), lactate dehydrogenase (LDH), prothrombin time (PT), activated partial thromboplastin time (APTT), international normalized ratio (INR), creatine kinase (CK), interleukin-6 (IL-6), lactate (Lac), and blood ammonia (NH_3_) in both initial and peak measurements during hospitalization (all *p* < 0.05). In contrast, arterial blood pH was significantly lower (*p* < 0.05). Additionally, peak values of creatine kinase-MB (CK-MB) and blood glucose (GLU) were significantly higher in the Non-Survivors (*p* < 0.05). No other laboratory parameters showed statistically significant differences (Table 3).

ALT and AST levels began to rise within 24 h after ingestion of *A. exitialis*, with a rapid, sharp increase in the non-survivors, contrasted with a mild elevation in the survivors. ALT and AST subsequently declined and returned to normal by day 4 in both groups (Figure 2A,B). In the non-survivors, TBIL increased markedly from day 4 post-ingestion, whereas no such elevation was observed in the survivors (Figure 2C). On day 4, bilirubin–enzyme dissociation (also known as enzyme–bilirubin dissociation) was observed in the non-survivors, characterized by markedly elevated bilirubin concentrations with disproportionately low aminotransferase activities. Blood urea nitrogen (BUN) and serum creatinine (Cr) rose significantly on day 2 post-ingestion and then declined; however, in the survivors, both indices increased again rapidly after day 9 (Figure 2D,E). Hematological parameters were also evaluated: leukocyte counts, platelet counts, red blood cell counts, and hemoglobin concentrations showed time-dependent declines throughout hospitalization in patients of both sexes (Figure 3A–F).

### 2.3. Amatoxin Detection

Results from toxin analysis of *A. exitialis* mushroom samples are summarized in Figure 4. All samples contained β-amanitin (β-AMA), α-amanitin (α-AMA), phallacidin (PCD), and phallisin (PSC), though concentrations varied across specimens. The phallotoxin-to-amatoxin ratio ranged from 0.67 to 1.59, indicating that total phallotoxin content generally exceeded total amatoxin content. Within phallotoxins, PSC predominated over PCD, whereas within amatoxins, α-AMA was more abundant than β-AMA.

All four toxins(α-AMA, β-AMA, PCD, and PSC)were detected in urine samples. Detection rates were 31.67% for α-AMA (1.1–72.6 ng/mL; 26–110 h; Figure 5A), 5.00% for β-AMA (1.8–8.3 ng/mL; 44–93 h; Figure 5B), 38.33% for PCD (1.1–16.1 ng/mL; 26–96 h; Figure 5C), and 49.17% for PSC (1.3–21.3 ng/mL; 26–110 h; Figure 5D). In individual patients, longitudinal measurements showed a gradual decline in urinary toxin concentrations over time, consistent with systemic clearance (Figure 5A,C,D). In blood, only three samples tested positive for α-AMA at 26 h (2.3 ng/mL), 51 h (2.1 ng/mL), and 88 h (7.2 ng/mL) post-ingestion.

As shown in Figure 6, Spearman correlation analyses were performed between the time-weighted urinary toxin exposure metric (concentration × time, C × t) and key biochemical indicators for α-AMA, β-AMA, PCD, and PSC; the results showed that α-AMA was significantly positively correlated with TBIL, AST, and Cr (r = 0.396–0.449, *p* < 0.05), PCD was positively correlated with TBIL, ALT, AST, APTT, PT, INR, CK, CK-MB, BUN, and Cr (r = 0.350–0.624, *p* < 0.05), and PSC exhibited positive correlations with TBIL, ALT, AST, BUN, and Cr (r = 0.159–0.590, *p* < 0.05).

### 2.4. Pathological Feature

Autopsy of a 3.5-year-old female patient revealed marked cerebral and cerebellar edema with congestion (Figure 7A,B), accompanied by multiple visceral petechial hemorrhages (Figure 7C–J). Hemorrhagic ascites and bilateral hemorrhagic pleural effusions were present in the abdominal and thoracic cavities (Figure 7K). The large intestine contained numerous nodular, black, hard fecal masses (Figure 7L).

Microscopic examination revealed cerebral edema (Figure 8A), prominent centrilobular necrosis with intrahepatic cholestasis and fatty degeneration in the liver (Figure 8B), granular degeneration with swelling of tubular epithelial cells and increased cytoplasmic eosinophilia in the renal tubules (Figure 8C), and alveolar hemorrhage accompanied by pulmonary edema in the lungs (Figure 8D). Widespread patchy hemorrhages involved the heart (interventricular septum and left ventricular tendons), gastrointestinal mucosa, gallbladder, and spleen (Figure 8E–I). Additionally, submucosal vessels of the stomach and small intestine showed endothelial necrosis and detachment.

### 2.5. Clinical Treatment

In this study, therapeutic approaches included supportive care, pharmacotherapy, and extracorporeal blood purification (Table 4); six patients received supportive care alone, with a mortality of 16.7% (1/6); nineteen patients were treated with N-acetylcysteine, benzylpenicillin, and silybin (NAC + PEN + SIL), among whom mortality was 15.8% (3/19); three patients received a dual regimen of NAC + PEG, with a mortality of 33.3% (1/3); two patients underwent combined hemoperfusion, plasmapheresis, and continuous venovenous hemodiafiltration (HP + PE + CVVHDF), with a mortality of 50.0% (1/2); eight patients received PE + CVVHDF, with a mortality of 37.5% (3/8); and one patient was treated with PE + CVVHDF plus a dual plasma molecular adsorption (DPMAS) and survived (0/1).

## 3. Discussion

Currently, more than 20 mushroom species containing amatoxins have been documented in China, primarily within the genera *Amanita*, *Galerina*, and *Lepiota*. *Galerina* and *Lepiota* predominantly contain amatoxins, whereas *Amanita* species harbor amatoxins as well as phallotoxins and virotoxins. Notably, over 70% of amatoxin-related mushroom poisonings in China are attributed to *Amanita* species [14]. Among these, *A. exitialis* is the most frequently implicated species. From 2000 to 2022, 73 poisoning events involving *A. exitialis* were reported worldwide, 72 of which occurred in China. These events involved 257 patients and resulted in 74 deaths, accounting for a case fatality rate of 28.8%, with a mean of 3.5 individuals poisoned and one death per event [15]. *A. exitialis* is primarily distributed in southern and southwestern China, often flourishing in forests dominated by Fagaceae plants. In regions where these conditions are met, it is essential to enhance the prevention and control measures against mushroom poisoning. In the present study, the ingested mushroom was definitively identified as *A. exitialis* by molecular identification.

To date, at least eight amatoxins have been detected in *A. exitialis*, with α-amanitin(α-AMA), β-amanitin(β-AMA), phallacidin(PCD), and phallisacin(PSC) reported as the most abundant [16]. In 2018, Sun quantified four amatoxins in *A. exitialis* using UPLC-ESI-MS and reported the concentration order α-AMA > β-AMA > PCD > γ-amanitin(γ-AMA) [10]. Consistently, our previous HPLC-based analysis identified α-AMA, β-AMA, PCD, and PSC across developmental stages of *A. exitialis* [17]. Hu further demonstrated tissue-specific distribution, with the highest toxin levels in the cap, followed by the stipe and the volva [18]. Prior studies have also shown that the composition and concentration of amatoxins and phallotoxins vary across species, geographic regions, growth stages, and tissue compartments of the same species [19]. Moreover, the total phallotoxin content in *A. exitialis* has been reported to be approximately 0.5–1.5 times that of amatoxins [17,20,21]. In our study, four toxins—α-AMA, β-AMA, PCD, and PSC—were detected in the cap, and the phallotoxin-to-amatoxin ratio ranged from 0.67–1.59, in line with prior literature.

Further testing revealed that toxins were more frequently detected in urine than in blood, with phallotoxins being more prevalent than amatoxins. The detection frequency was ranked as PSC > PCD > α-AMA > β-AMA. Only three blood samples tested positive for α-AMA, all from patients who developed organ injury (one of which was fatal due to liver failure). Although previous reports cite α-AMA detection windows of less than 36 h in blood and less than 72 h in urine [22], we detected it up to 88 h in blood and 110 h in urine. Evidence regarding phallotoxin absorption has been inconsistent: early mouse studies suggested poor gastrointestinal uptake due to the lack of specific transporters [23], whereas subsequent dog experiments and human case series detected phallotoxins in blood and/or urine [24,25]. Our findings align with the latter, demonstrating the presence of both amatoxins and phallotoxins in patient urine and supporting the gastrointestinal absorption of phallotoxins. Given their longer detection window and higher positivity in urine—particularly PSC and PCD—phallotoxins are promising diagnostic biomarkers. Therefore, we recommend extending urine and blood sampling beyond conventional timeframes to enhance diagnostic yield and enable timely, precise treatment. In addition, we conducted a correlation analysis between the product of peak urinary toxin concentration and sampling time, as well as laboratory parameters. The results indicated significant positive correlations between toxin accumulation and levels of total bilirubin TBIL, AST, ALT, BUN, Cr, CK, CK-MB, PT, APTT, and INR. Specifically, higher levels of toxin accumulation in the body were associated with more severe symptoms of multiorgan dysfunction, with particularly notable damage to the liver and kidneys. This emphasizes the critical importance of timely and accurate toxin detection in the early stages of poisoning for assessing disease severity and guiding clinical treatment.

The currently recognized mechanism holds that amatoxins are absorbed from the intestine, enter the portal circulation, and are taken up by hepatocytes via NTCP and OATP1B1/1B3, primarily causing liver injury [26,27]. As illness advances, secondary complications—renal injury, myocardial dysfunction, and coagulation disorders—can emerge [26,27]. Consistent with transporter-mediated susceptibility, Gong showed that cytotoxicity correlates with OATP1B3 expression across human organ–derived cell lines [28]. In our cohort, laboratory data confirmed hepatic, renal, and cardiac injuries with coagulation dysfunction; autopsy likewise demonstrated marked hepatocellular damage alongside renal and myocardial injuries, cerebral edema, gastrointestinal mucosal shedding, and multifocal hemorrhages. In this study, the non-survivor group exhibited significantly worse deterioration across all indicators compared to the survivor group. This disparity reflects differences in the rates of disease progression between the two groups and suggests that non-survivors are at a higher risk of treatment complications and more likely to experience adverse outcomes. In contrast, the survivors exhibited significantly elevated kidney function indicators in the later stages. This progression suggests that severe kidney damage occurs in the advanced stages. Moreover, this trend highlights the importance for physicians to closely monitor changes in renal function parameters during treatment and to promptly adjust therapeutic strategies to promote renal recovery. These findings offer critical guidance for clinicians in the early identification of high-risk patients, the formulation of precise treatment strategies, and the improvement of patient outcomes.

Notably, deceased patients developed enzyme–bilirubin dissociation from day 4 post-ingestion, signaling fulminant hepatic failure that progressed to multiple organ dysfunction syndrome—a stage associated with 30–60% mortality [29]. This pattern is a critical prognostic marker of poor outcome; liver transplantation is the only potentially life-saving therapy at this stage [30]. Gastrointestinal injury in the deceased—especially pronounced abdominal distension—mirrored the case reported by Meng [11], where paralytic ileus ensued and was fatal. When severe abdominal distension is present, gastrointestinal decontamination (e.g., gastric lavage, cathartics) is not recommended to avoid aggravating intestinal injury. As reported by Miranda Visser et al., hematotoxicity was observed in our cohort, evidenced by reductions in hemoglobin, red blood cells, platelets, and white blood cells; importantly, this hematotoxicity appears secondary to hepatic injury and coagulopathy [31]. Despite a lack of robust evidence, symptomatic supportive care—including plasma exchange (PE), hemoperfusion (HP), and continuous venovenous hemodiafiltration (CVVHDF)—remains the mainstay treatment for amatoxin-related multiorgan toxicity (hepatic, renal, cardiac, gastrointestinal, and hematologic) [32,33,34,35,36,37,38].

## 4. Conclusions

*A. exitialis* mainly contains α-AMA, β-AMA, PCD, and PSC, which could be detected up to 88 h in blood and 110 h in patients’ urine. Amatoxins and phallotoxins are promising diagnostic biomarkers for mushroom poisoning. Liver is the main damaged organ of *A. exitialis* poisoning, which also shows multiorgan toxicity (renal, cardiac, gastrointestinal, and hematologic). Enzyme–bilirubin dissociation and abdominal distension were critical prognostic markers of poor outcome. And current management of amatoxin poisoning remains largely supportive.

*A. exitialis* mainly contains α-AMA, β-AMA, PCD, and PSC, which were detectable up to 88 h in blood and 110 h in urine in our cohort, indicating that amatoxins are practical diagnostic biomarkers; the liver is the primary target with subsequent multiorgan toxicity involving renal, cardiac, gastrointestinal, and hematologic systems; enzyme–bilirubin dissociation is a critical prognostic indicator that signals impending fulminant hepatic failure and poor outcomes; current management remains largely supportive (e.g., plasma exchange, hemoperfusion, CVVHDF) with liver transplantation as the only potentially life-saving option for fulminant hepatic failure.

## 5. Materials and Methods

### 5.1. Patients and Study Design

We conducted a retrospective analysis of *A. exitialis* poisoning cases treated at Chuxiong Yi Autonomous Prefecture People’s Hospital from 2019 to 2024. Case confirmation was based on molecular identification of the ingested mushrooms. Exclusion criteria were a history of liver, kidney, cardiac, or coagulation disorders, or excessive alcohol consumption at the time of ingestion. See Figure 9 for details on the method flowchart. The study complied with the Declaration of Helsinki and was approved by the Medical Ethics Committee of Chuxiong Yi Autonomous Prefecture People’s Hospital (Approval No. 2022-40). The analysis of autopsies on deceased children has been conducted with informed consent.

### 5.2. Data Collection and Definitions

Data for gender, age, time of poisoning, incubation period, time from ingestion to hospital admission, duration of hospitalization, complications, laboratory findings, autopsy pathology, and clinical treatment were collected from the medical records of the patients. Clinical parameters were defined as follows: liver injury as alanine aminotransferase (ALT) > 50 U/L and/or aspartate aminotransferase (AST) > 50 U/L; renal injury as creatinine (Cr) > 106 μmol/L and/or blood urea nitrogen (BUN) > 6.1 mmol/L; myocardial injury as creatine kinase (CK) > 153 U/L and/or creatine kinase isoenzymes (CK-MB) > 24 U/L; coagulation dysfunction as international normalized ratio (INR) > 1.5; thrombocytopenia as platelet count < 125 × 10^9^/L; lactic acidosis as lactate > 2.2 mmol/L; and anemia as hemoglobin < 120 g/L for females or <110 g/L for males.

### 5.3. Grouping

Patients were classified into survivor and non-survivor groups based on clinical outcomes. We compared laboratory indicators between groups and evaluated the temporal trajectories of hepatic function (AST, ALT, TBIL) and renal function (BUN, Cr) following poisoning.

### 5.4. Identification of Toxic Mushrooms

Molecular identification of suspected mushroom specimens was performed using the internal transcribed spacer (ITS) region, as described previously [14]. The ITS region was amplified with primers ITS5 and ITS4, and PCR products were sequenced by KingMed Biotechnology Co., Ltd. (Kunming, China). Sequence homology was verified using BLAST (version 2.14.0) in the NCBI database, and confirmed sequences were deposited in GenBank to obtain accession numbers.

### 5.5. Toxin Detection

Toxins were detected in mushroom specimens, patient blood, and urine using ultra-high-performance liquid chromatography–tandem mass spectrometry (UHPLC–MS/MS), as previously described [39]. The toxin detection utilized the external standard method, employing α-AMA, β-AMA, and PCD reference materials (provided by ENZO Life Sciences., Farmingdale, NY, USA) and PSC reference material (provided by Fuzhou Qinpeng Biotechnology Co., Fuzhou City, China). The detection limits, quantification limits, standard curves, and correlation coefficients for blood and urine samples are detailed in Table 5.

### 5.6. Statistical Analysis

Categorical variables were summarized as frequencies (*n*) and percentages (%), whereas continuous variables with approximately normal distributions were expressed as mean ± standard deviation and compared using one-way analysis of variance (ANOVA); skewed continuous variables were presented as median and interquartile range (IQR) and analyzed using the Mann–Whitney U test. Correlation analyses were conducted using Pearson’s method for parametric data and Spearman’s method for non-parametric data. All statistical analyses were performed using GraphPad Prism, version 10.0 (GraphPad Software, San Diego, CA, USA), and a *p*-value < 0.05 was considered statistically significant.

## Figures and Tables

**Figure 1 toxins-17-00576-f001:**
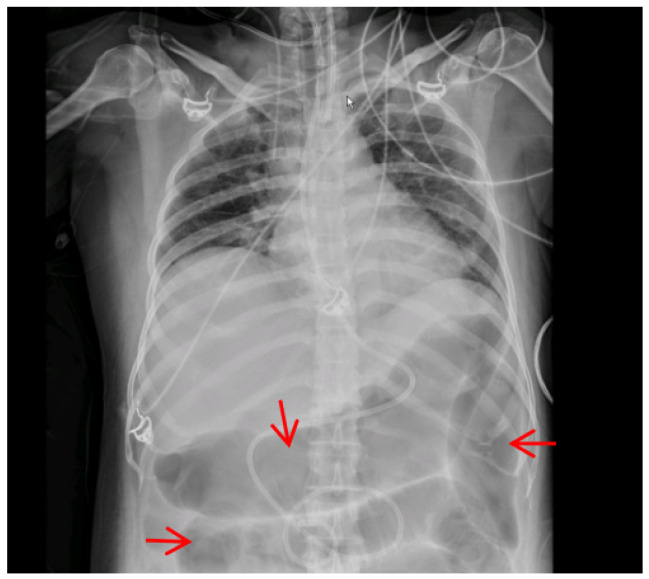
Abdominal anteroposterior radiograph showing an intestinal gas shadow, indicated by a red arrow.

**Figure 2 toxins-17-00576-f002:**
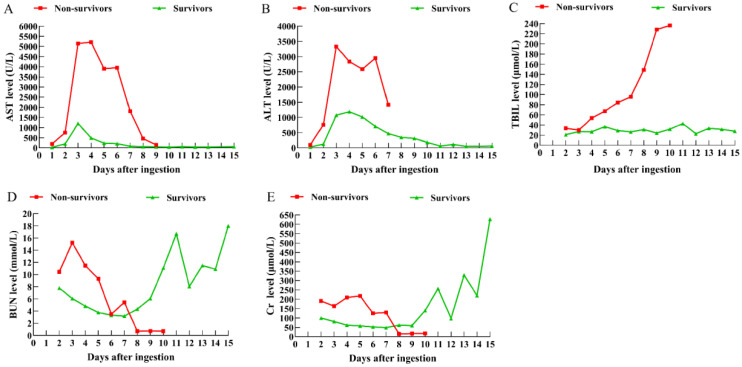
Dynamic changes in liver and kidney function markers in patients after *A. exitialis* ingestion. (**A**) AST; (**B**) ALT; (**C**) TBIL; (**D**) BUN; (**E**) Cr. The error bars are not displayed because the individual differences within the sample are significant.

**Figure 3 toxins-17-00576-f003:**
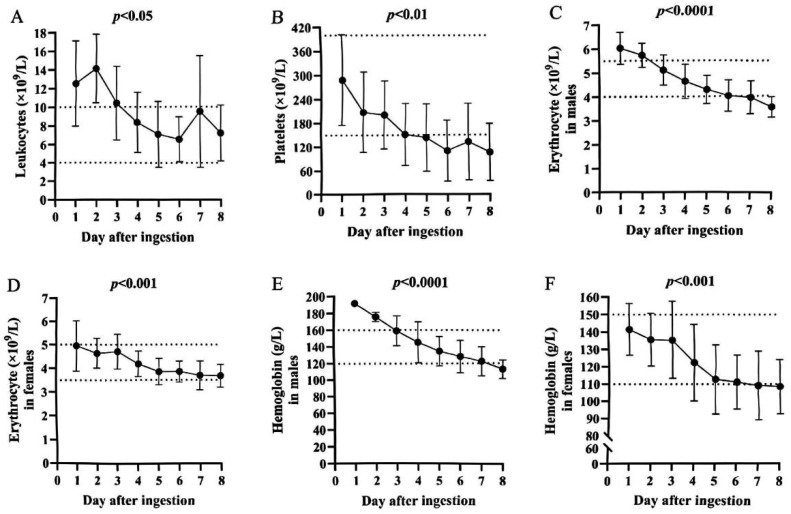
Hematological parameter changes following *A. exitialis* ingestion: (**A**) Leukocytes; (**B**) platelets; (**C**) Erythrocyte in males; (**D**) Erythrocyte in females; (**E**) Hemoglobin in males; (**F**) Hemoglobin in females. Dashed lines indicate reference ranges.

**Figure 4 toxins-17-00576-f004:**
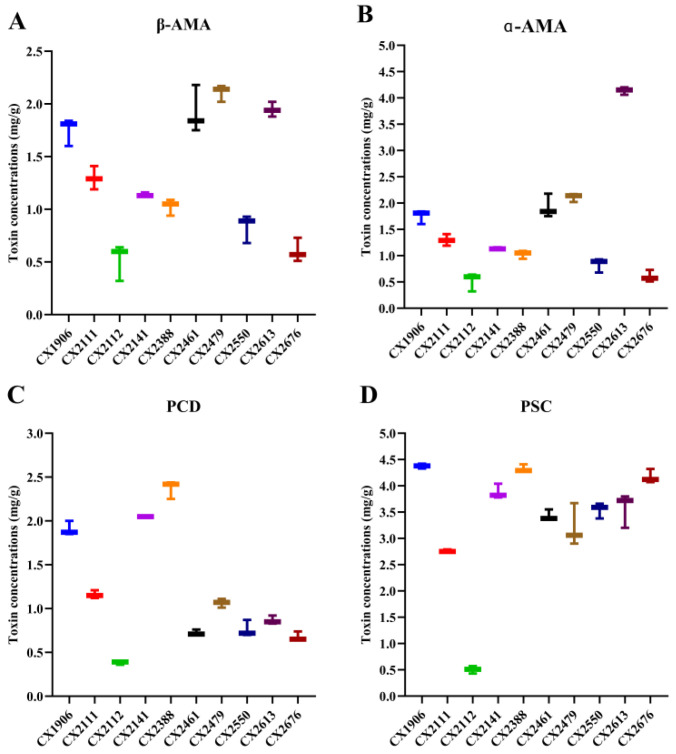
Toxin concentrations of *A. exitialis* samples (mg/g). Same color means same sample. (**A**) α-amanitin (α-AMA); (**B**) β-amanitin (β-AMA); (**C**) phallacidin (PCD); (**D**) phalloidin (PSC).

**Figure 5 toxins-17-00576-f005:**
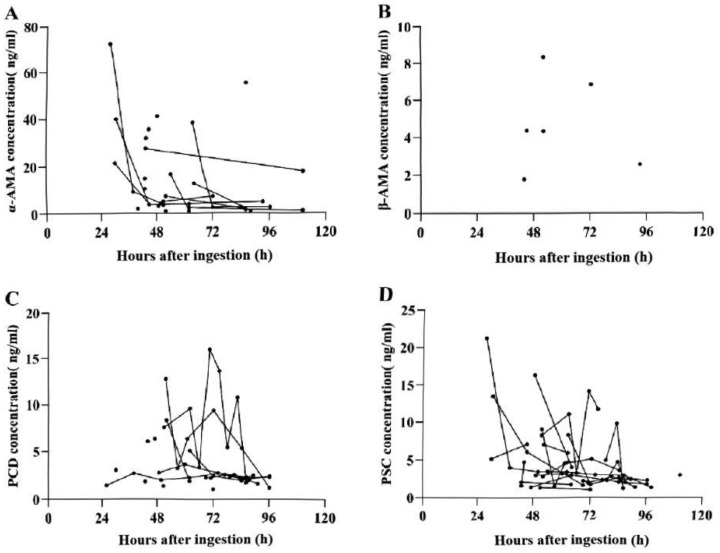
Scatter plots of urinary toxin concentrations in patients with *A. exitialis* poisoning: (**A**) α-AMA; (**B**) β-AMA; (**C**) PCD; (**D**) PSC. The black dots in the figure represent the concentration of toxins detected in the urine, and the test data connected by the same line pertain to the same patient.

**Figure 6 toxins-17-00576-f006:**
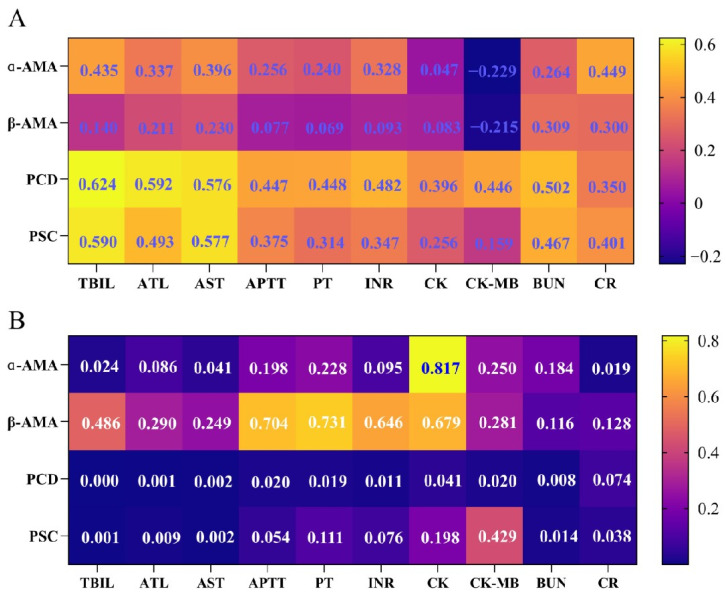
Heatmaps of Spearman correlation between the time-weighted urinary toxin exposure metric and key biochemical indicators: (**A**) Spearman correlation coefficients; (**B**) *p* values from Spearman correlation tests.

**Figure 7 toxins-17-00576-f007:**
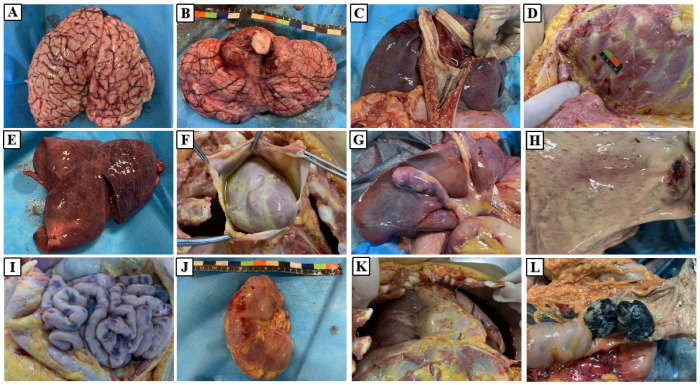
Gross anatomical findings of the patient’s dissected organs: (**A**) Brain; (**B**) Cerebellum; (**C**) Trachea; (**D**) Intercostal muscles; (**E**) Lungs; (**F**) Heart; (**G**) Liver and gallbladder; (**H**) Gastric mucosa; (**I**) Small intestine; (**J**) Left kidney; (**K**) Left and right thoracic cavities; (**L**) Large intestine.

**Figure 8 toxins-17-00576-f008:**
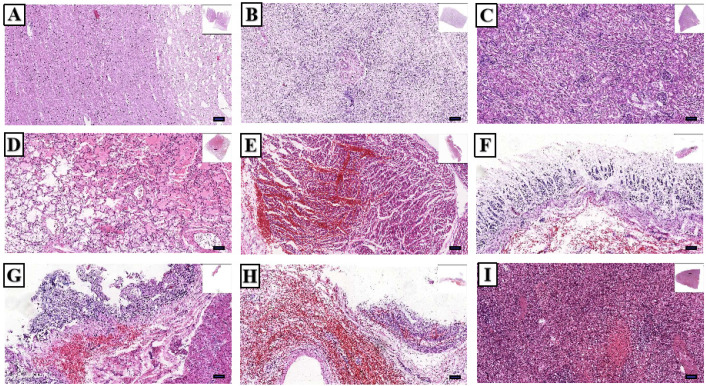
Histopathological findings of dissected organs from a fatal *A.exitialis* poisoning case: (**A**) Brain; (**B**) Liver; (**C**) Kidney; (**D**) Lung; (**E**) Heart; (**F**) Stomach; (**G**) Small intestine; (**H**) Gallbladder; (**I**) Spleen. (×100, scale =100 µm).

**Figure 9 toxins-17-00576-f009:**
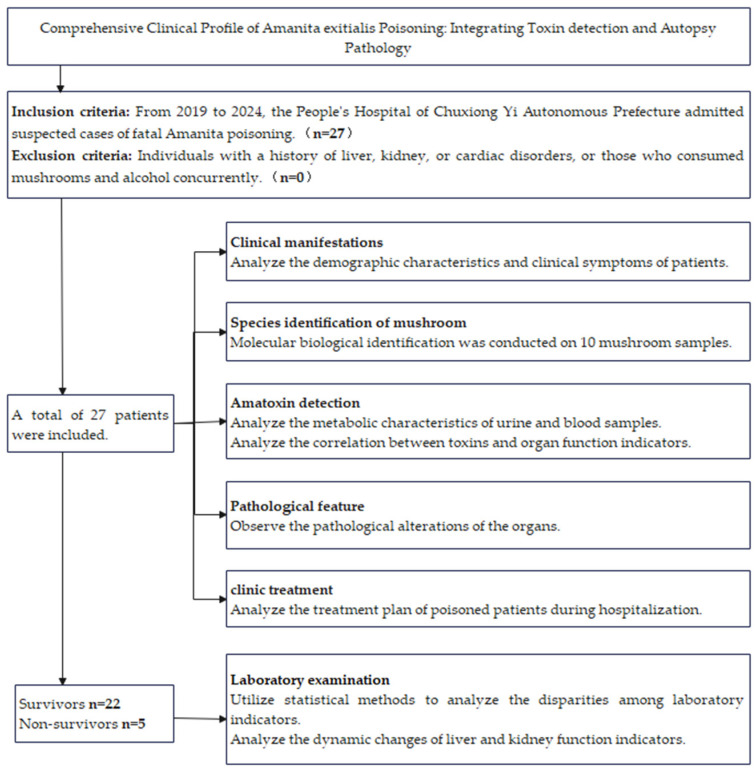
Flowchart of this study method.

**Table 1 toxins-17-00576-t001:** GenBank accession numbers for ITS sequences from *A. exitialis* in this study.

Specimen Number	ITS GenBank Accession Number	Location
CX2550	PQ394610	Chuxiong City
CX1906	PQ394599	Yao’an County
CX2676	PQ394605	Wuding County
CX2111	PQ394600	Yongren County
CX2388	PQ394602	Dayao County
CX2613	PQ394604	Wuding County
CX2479	PQ394609	Wuding County
CX2141	PQ394601	Yuanmou County
CX2461	PQ394603	Wuding County
CX2112	PQ394608	Chuxiong City

**Table 2 toxins-17-00576-t002:** General characteristics of patients with *A. exitialis* poisoning.

Characteristics	Total (*n* = 27)	Survivors (*n* = 22)	Non-Survivors (*n* = 5)	*p* Value
Gender, *n* (%)				
Male	13 (48)	12 (92)	1 (8)	0.3259
age (year)	46.8 ± 22.1	46.5 ± 21.6	48.2 ± 27.0	0.7590
Month, *n* (%)				
June	13 (48)	11 (85)	2 (15)	/
July	10 (37)	7 (70)	3 (30)	/
August	4 (15)	4 (100)	0 (0)	/
latency (h)	12.0 ± 5.0	12.5 ± 5.2	10.0 ± 3.9	0.3240
Ingestion to hospital admission (h)	45.9 ± 8.6	45.3 ± 8.9	48.7 ± 7.8	0.4400
Hospital duration (days)	8.2 ± 5.8	8.9 ± 6.2	5.0 ± 2.3	0.1770
Complications (%)				
Gastrointestinal Symptoms	27 (100)	22 (81)	5 (19)	1.0000
liver injury	21 (78)	16 (76)	5 (24)	0.5550
Kidney injury	17 (63)	14 (82)	3 (17)	1.0000
Myocardial injury	24 (89)	19 (79)	5 (21)	1.0000
Coagulopathy dysfunction	10 (37)	6 (60)	4 (40)	0.0470
Thrombocytopenia	15 (56)	11 (73)	4 (27)	0.3420
Hyperlactatemia	8 (30)	4 (50)	4 (50)	0.0172
Anemia	13 (48)	10 (77)	3 (23)	0.6480
Paralytic ileus	1 (4)	0 (0)	1 (100)	0.1852

**Table 3 toxins-17-00576-t003:** Initial and Peak laboratory values in non-survivors versus survivors.

Laboratory Finding	Reference Range	Survivors (*n* = 22)	Non-Survivors (*n* = 5)	*p* Value
Initial TBIL (umol/L)	3–22	23.3 ± 14.7	51.4 ± 10.0	0.0004
Peak TBIL (umol/L)	3–22	36.7 ± 35.0	112.6 ± 71.0	0.0037
Initial ALT (U/L)	<50	32 (27, 860)	1842 (733, 4963)	0.0164
Peak ALT (U/L)	<50	85 (38, 1685)	2008 (1925, 7640)	0.0021
Initial AST (U/L)	15–46	43 (29, 1333)	1900 (501, 7110)	0.0243
Peak AST (U/L)	15–46	78 (41, 2098)	4414 (2792, 9179)	0.0028
Initial LDH (U/L)	313–618	266 (204, 2140)	5700 (2302, 20,597)	0.0104
Peak LDH (U/L)	313–618	421 (256, 3515)	9380 (7322, 29,271)	0.0027
Initial Cr (umol/L)	62–106	68.3 (41.0, 143.7)	67.4 (44.7, 361.0)	0.9357
Peak Cr (umol/L)	62–106	81.5 (56.9, 143.7)	144.4 (65.6, 437.0)	0.2316
Initial BUN (mmol/L)	2.5–6.1	6.5 ± 2.6	9.7 ± 9.1	0.7932
Peak BUN (mmol/L)	2.5–6.1	8.7 ± 5.5	12.2 ± 8.4	0.4848
Initial PT (S)	11–13	12.1 (11.4, 12.9)	25.8 (17.9, 45.4)	0.0025
Peak PT (S)	11–13	14.0 (11.9, 26.0)	90.0 (37.1, 103.6)	0.0122
Initial APTT (S)	23–37	27.2 (24.9, 29.9)	53.7 (32.3, 88.4)	0.0104
Peak APTT (S)	23–37	34.3 (26.1, 92.1)	120.0 (73.5, 120.0)	0.0264
Initial INR	0.8–1.5	1.05 (0.99, 1.12)	2.30 (1.58, 4.16)	0.003
Peak INR	0.8–1.5	1.10 (0.99, 2.34)	9.26 (3.50, 9.99)	0.0045
Initial CK (U/L)	30–153	94 (66, 186)	214 (151, 1600)	0.0341
Peak CK (U/L)	30–153	166 (101, 269)	394 (218, 13,091)	0.0126
Initial CK-MB (U/L)	0–24	27.7 (15.7, 40.0)	36.8 (23.1, 132.1)	0.3512
Peak CK-MB (U/L)	0–24	38 (25, 57)	67 (23, 431)	0.0043
Initial IL-6 (pg/mL)	0–7	5.0 (1.5, 19.3)	88.3 (24.4, 169.0)	0.0036
Peak IL-6 (pg/mL)	0–7	17.8 (5.0, 49.5)	210.5 (174.5, 2616.0)	0.0002
Initial GLU (mmol/L)	3.5–6.1	6.8 (5.0, 8.1)	6.2 (5.7, 8.7)	0.8867
Peak GLU (mmol/L)	3.5–6.1	8.2 (7.0, 10.1)	11.0 (10.6, 18.2)	0.0033
Initial Lac (mmol/L)	0.5–2.2	1.10 (0.90, 1.35)	4.70 (2.35, 5.00)	0.0047
Peak Lac (mmol/L)	0.5–2.2	1.9 (1.3, 3.4)	15.0 (7.8, 16.0)	0.0010
Initial Amon (umol/L)	9–30	30.7 ± 16.7	97.3 ± 40.7	0.0011
Peak Amon (umol/L)	9–30	53.4 ± 29.8	186.2 ± 86.7	0.0013

Abbreviations: ALT, alanine transaminase; Amon, ammonia; APTT, activated partial thromboplastin time; AST, aspartate aminotransferase; BUN, blood urea nitrogen; CK, creatine kinase; CK-MB, creatine kinase myocardial band; Cr, creatinine; GLU, glucose; IL-6, interleukin-6; INR, international normalized ratio; Lac, lactic acid; LDH, lactate dehydrogenase; PT, prothrombin time; TBIL, total bilirubin.

**Table 4 toxins-17-00576-t004:** Treatment modalities and outcomes in patients with *A. exitialis* poisoning.

Treatment Modality	Number of Cases	Number of Deaths	Mortality Rate (%)
Supportive therapy	6	1	16.7%
Pharmacological therapy			
NAC + SIL	5	1	20%
NAC + PEN	3	1	33.3%
NAC + SIL + PEN	19	3	15.8%
Activated charcoal	23	5	21.7%
Ganoderma decoction	11	2	18.2%
Extracorporeal blood purification			
HP	5	0	0%
CVVHDF	1	0	0%
HP + CVVHDF	4	0	0%
PE + CVVHDF	8	3	37.5%
HP + PE + CVVHDF	2	1	50%
PE + CVVHDF + DPMAS	1	0	0%

Abbreviations: NAC, N-acetylcysteine; SIL, silibinin; PEN, benzylpenicillin; HP, hemoperfusion; CVVHDF, continuous venovenous hemodiafiltration; PE, plasmapheresis; DPMAS, dual plasma molecular adsorption.

**Table 5 toxins-17-00576-t005:** Detection limits, quantification limits, linear relationships, and correlation coefficients of toxins in blood and urine.

Toxin Type	Linear Relationship	Correlation Coefficients	Detection Limits (ng/mL)	Quantification Limits (ng/mL)
α-AMA	y = 646.32x + 1890.00	R^2^ = 0.9990	1.00	1.00
β-AMA	y = 804.66x − 1586.30	R^2^ = 0.9971	1.00	1.00
PCD	y = 3458.60x − 1086.00	R^2^ = 0.9953	0.01	1.00
PSC	y = 1886.10x − 621.93	R^2^ = 0.9998	0.01	1.00

## Data Availability

The original contributions presented in this study are included in the article. Further inquiries can be directed to the corresponding authors.

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
