# Peer review of "Comprehensive Clinical Profile of Amanita exitialis Poisoning: Integrating Toxin Detection and Autopsy Pathology"

_toxins, 2025, doi:10.3390/toxins17120576_

Round 1
Reviewer 1 Report
Comments and Suggestions for Authors
This interesting paper describes a retrospective analysis on the outcome of A. exitialis poisonings in China. The results are interesting as report on A. exitialis poisonings are limited and the authors performed a detailed analysis on a high number of parameters. While the results are very relevant, there are several aspects of the manuscript that require clarification.
Specific comments:
- In the abstract it is indicated that ‘Fatal cases showed bile-enzyme dissociation at day 4 post-ingestion’. This mechanism is new to me and maybe also to other readers. I would suggest to rephrase this sentence.
- The abstracts end with the statement that ‘Urinary toxin testing should be used to aid the diagnosis and management of poisoning by amatoxin-containing mushrooms’. Although this statement is consistent with the contents of the manuscript, no information is presented in the abstract to support this statement. In addition, a conclusion appears to be missing from the abstract.
- In the results the authors indicate that all collected mushrooms were identified as A. exitialis. From the description it is currently not clear whether this were all mushrooms and what the relation of these mushrooms was with the cases.
- Interesting fact from the laboratory measurements was that the non-survivors group were already worse compared to the measurements in the survivors group. I think this fact should also be discussed by the authors.
- For the kidney laboratory markers a increased rather than a decrease was observed after day 9 for survivors. This is opposite to my expectations from literature. I think this should be discussed by the authors.
- Changes in liver- and kidney function markers are expressed in Figure 2. Error bars appear to be missing from these figures. In addition, average admission time of the non-survivors was 5 days, but data are presented up to ten days. This may appear a bit confusing. This is probably due to statistics, but maybe a statement or clarification could be included.
- Previous studies have indicated that changes in the hematological parameters were related to the severity of the poisoning. Did the authors find a similar relationship in this study?
- The authors measured α-AMA, β-AMA, PCD, and PSC in urine. They showed that there was a correlation with various laboratory measurements, but the most important correlation appears to remain unanswered. Did the authors also find a relationship with the outcome of the poisoning?
- In Figure 5 not all values are readable. Especially, the combination of white and yellow is not ideal. I would suggest to use another font color.
- In the clinical treatment paragraph several treatments (pharmacological and purification therapies) appear to be combined. As currently described, it is not at the not clear from if this was the case. In addition, it would be interesting to know on which basis treatment was selected. Was this based on the severity of the poisoning at presentation, on the complications, or on other basis?
- Small detail, there are a number of typographical errors throughout the manuscript.
Reviewer 2 Report
Comments and Suggestions for Authors
Dear Authors and Editors,
This extensive and interesting work on fungal toxins and their impact on the biological imbalances in the human body is a truly comprehensive work.
It should be emphasized that the manuscript includes a detailed description of toxicity symptoms, both morphological and physiological, including pathogen and gene analyses. In its current form, it is practically ready for publication, as it contains a wealth of biological data: content, pathology, and graphics. Minor errors have been highlighted in the PDF file.
Sincerely,
Reviewer

Reviewer 3 Report
Comments and Suggestions for Authors
Please consider the following comments while you revise the content of this paper.
1- Please mention that the biopsy images mentioned that in Figure 6 were for a dead person or they were prepared during a surgery?
2- In the title of this manuscript, please make the scientific name of Amanita exitialis italic. In other parts of paper, similar strategy should also be followed. For the second time mentioning the name of Amanita exitialis, the author should discuss it as: A. exitialis
3- Please clarify that the share of figures in figure 6 was ethical and it way approved by your department. Please supplement a copy of ethical approval for this study along with a formal letter from the hospital and lab that these biopsies were taken from dead or live patients.
4- Please discuss the novelty of this study in the introduction section.
5- Please add a flowchart to the methodology of your paper to summarize the experimental sections discussed there.
6- Please use clustered heatmaps to discuss your data in figure 5.
7- Please use a box plot to discuss mean and SD data represented in table 4
8- The study addresses an important and underexplored aspect of Amanita exitialis poisoning-integrating toxin quantification with autopsy pathology. This combined clinical–forensic perspective is valuable. However, the authors should more clearly articulate how their findings advance current understanding beyond prior amatoxin toxicity studies or other Amanita species reports. Highlighting this in the introduction and discussion would strengthen the paper’s novelty.
9-The methods are comprehensive but could benefit from greater detail regarding the UHPLC–MS/MS toxin quantification. Please specify detection limits (LOD/LOQ), calibration curves, internal standards used, and quality control procedures to ensure analytical reliability. This information is critical for readers seeking to reproduce or compare the toxin quantification results.
10- While the statistical methods are appropriate, the manuscript should clarify the sample sizes for each test and the handling of missing data (e.g., incomplete urine or blood samples). Moreover, correlation analyses between urinary toxin levels and biochemical indices would benefit from visualization (e.g., scatter plots with regression lines) and inclusion of correlation coefficients (r) and exact p-values in the results section.
11-The pathological descriptions are informative, but they would be significantly strengthened by including representative histopathological images (e.g., liver, kidney, and heart tissues) with scale bars and appropriate staining details. Additionally, consider providing a brief comparison between fatal and nonfatal cases to contextualize the severity of tissue damage
12-The correlation between urinary amatoxin concentration and organ injury markers is compelling. However, the discussion could be expanded to address potential diagnostic or prognostic applications-for instance, how early urinary toxin testing could influence treatment decisions (e.g., use of N-acetylcysteine, silibinin, or liver transplantation criteria). Including this clinical translation would enhance the impact of the study. A robust and comprehensive discussion should be add to the paper.
13- Please check all references to make sure problematic papers were not cited within your paper.
14- Add DOI identifiers to all cited references.
15- Add image and geographic distribution of Amanita exitialis in your location and other possible growing area to inform readers about the potential toxicity of this toxic fungus.
Round 2
Reviewer 3 Report
Comments and Suggestions for Authors
Thanks for your revision. I have no further comments.
Best regards